# ETGS: Explicit Thermodynamics Gaussian Splatting for Dynamic Thermal Reconstruction

**Zhongwen Wang[1], Han Ling [1], Weihao Zhang [1], Yinghui Sun [2], Quansen Sun [1*]**
[1]Nanjing University of Science and Technology, [2]Southeast University
{jankinwang, 321106010190, zhangweihao, sunquansen}@njust.edu.cn,
sunyh@seu.edu.cn

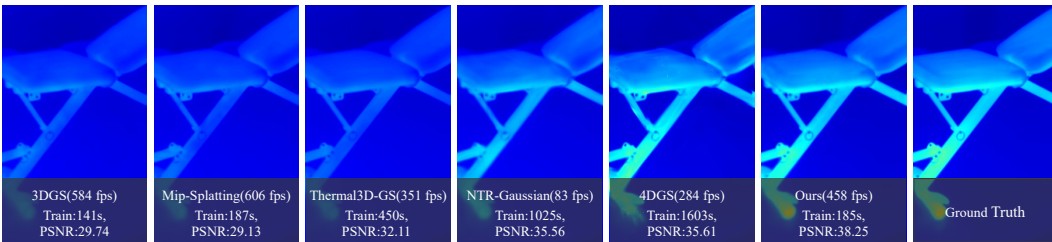

Figure 1: Our method achieves high-quality rendering of dynamic thermal scenes with efficiency comparable to static methods (Kerbl et al. (2023); Yu et al. (2024)). The key to this performance is the novel explicit modeling of dynamic thermal Gaussians based on thermodynamics, which significantly speeds up scene optimization and synthesis of new views, while achieving state-of-the-art quality.

## ABSTRACT

We propose ETGS, a method for reconstructing dynamic thermal scenes by embedding explicit thermodynamic modeling into 3D Gaussian Splatting. Each Gaussian is equipped with physically interpretable thermal parameters, and its thermodynamics evolution is described by a first-order heat-transfer ODE with an analytical closed-form solution. This formulation avoids numerical integration, enables efficient rendering at arbitrary timestamps, and naturally handles irregular sampling and out-of-order observations. We also introduce the Rapid Heat Dynamics (RHD) dataset, which provides millisecond-aligned RGB–IR image pairs covering typical thermal processes such as cooling, warming, heating, and heat transfer. Experiments on RHD show that ETGS captures rapid thermal dynamics more accurately than existing static and dynamic baselines, while maintaining training and rendering efficiency close to that of static 3DGS. Code and dataset are available at https://github.com/jankin-wang/ETGS

# 1 INTRODUCTION

Thermal imaging is a non-contact temperature measurement method that can directly capture thermal radiation signals from an object's surface and maintains stable operation even in complex environmental conditions such as low light levels (Wilson et al. (2023); Li et al. (2024a); Xu et al. (2025); Shin & Park (2025)). Compared to traditional visible light imaging, thermal imaging not only provides geometric structural information but also reflects the temperature distribution characteristics of an object, thus demonstrating unique advantages in 3D scene modeling and physical process analysis (Aibibu et al. (2025); Nowakowski & Kaczmarek (2025); Wang et al. (2025); Wu et al. (2020); Ramon et al. (2022)). Therefore, combining thermal imaging with 3D reconstruction

---

*Corresponding author.

to generate temperature scene models that evolve over time has become a research hotspot in recent years.

Neural Radiance Fields (NeRF) (Mildenhall et al. (2021)) uses implicit neural networks to model scenes, producing high-quality results in novel perspective synthesis tasks. In thermal scene modeling, Thermal-NeRF (Ye et al. (2024)) and ThermoNeRF (Hassan et al. (2024)) extend NeRF to IR images, achieving 3D temperature field reconstruction. However, NeRF's implicit representation suffers from geometric instability and inefficiencies in training and rendering (Wang et al. (2024); Li et al. (2023; 2024b); Korhonen et al. (2024)), making it difficult to implement in dynamic scenes. Unlike NeRF, 3D Gaussian Splatting (3DGS) (Kerbl et al. (2023)) uses explicit Gaussians to model scenes, and all parameters (position, size, orientation, color, etc.) can be learned, resulting in strong scalability and downstream compatibility (Fei et al. (2024); Charatan et al. (2024); Tang & Cham (2024)). Recently, methods such as Thermal3D-GS (Chen et al. (2024)) and TGA-GS (Zou et al. (2025)) have introduced 3DGS into thermal reconstruction tasks, achieving 3D modeling and perspective synthesis of thermal scenes. However, they are all limited to static scenarios and fail to capture the dynamic process of temperature change over time, so they have obvious limitations in thermodynamic analysis.

To address dynamics, some work has introduced temporal modeling on 3DGS. 4DGS (Wu et al. (2024)) incorporates temporal variables into the Gaussian representation, enabling it to reconstruct dynamic appearances, but does not consider thermophysical processes. ThermalGS (Liu et al. (2025)) attempts to drive the temporal evolution of the Gaussian through temporal embedding and semantic information, but is essentially still data-driven modeling and lacks thermodynamic consistency. NTR-Gaussian (Yang et al. (2025)) introduces thermodynamic equations into the Gaussian framework, but it relies on implicit neural networks and integral inference, which limits its training and rendering efficiency.

To address these limitations, we propose ETGS, a reconstruction method for dynamic thermal scenes based on explicit thermodynamic modeling. We directly introduce explicit thermal physical variables into the Gaussian representation and derive the closed-form solution of the ordinary differential equation (ODE) at arbitrary time, avoiding the overhead of numerical integration and naturally adapting to unequal sampling and disordered timestamps. By combining explicit thermodynamic modeling with the Gaussian rendering framework, the proposed method is close to static 3DGS in training and inference efficiency, while achieving significantly better performance than recent state-of-the-art baselines in dynamic thermal reconstruction tasks. In addition, we also constructed an RHD dataset that covers typical processes such as warming, cooling, heating and heat transfer, and provides millisecond timestamps and pixel-aligned RGB-IR data, thus laying the foundation for quantitative research on dynamic thermal scenes. Our contributions are summarized as follows:

- We propose a dynamic thermal scene reconstruction method based on explicit thermodynamic physics. We construct a reliable thermodynamic model using equivalent heat capacity, heat exchange coefficients, and heat source excitations. We derive closed-form solutions to ODEs at arbitrary time, enabling efficient rendering under unequal sampling and out-of-order sampling.

- We introduce the Rapid Heat Dynamics (RHD) dataset, a new dataset designed specifically for rapidly changing thermal dynamics. It covers the thermodynamic processes of warming, cooling, heating, and heat transfer, along with millisecond-accurate timestamps and pixel-aligned RGB data.

- Experimental results demonstrate that the proposed method reliably reconstructs rapidly changing thermal dynamics with comparable training and rendering efficiency to static 3DGS, and outperforms existing methods across various metrics.

## 2 RELATED WORK

### 2.1 3D THERMAL SCENE RECONSTRUCTION

3D thermal scene reconstruction aims to recover both scene geometry and thermal radiation, supporting applications such as energy efficiency monitoring, industrial inspection, and medical diagnosis (He et al. (2021); Glowacz (2021); Lahiri et al. (2012); Zhou et al. (2021)). Early methods

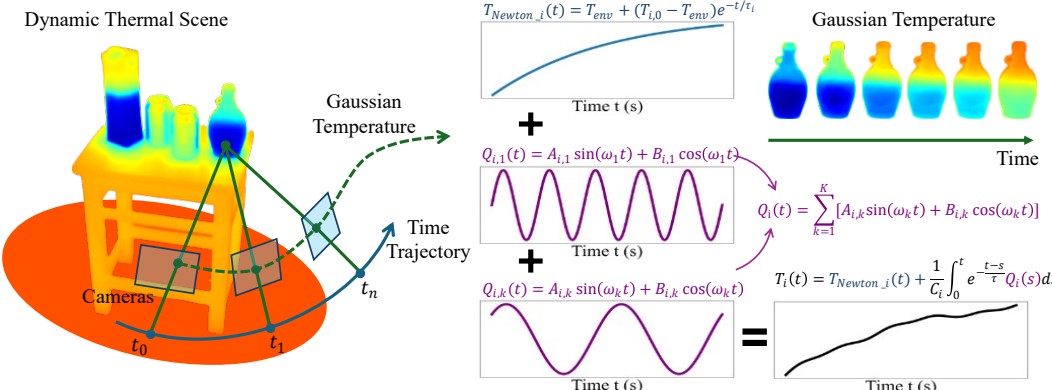

Figure 2: **Method Overview.** ETGS directly incorporates thermal physics modeling into the explicit Gaussian scene representation. The temperature of each Gaussian consists of two components: an exponential term ($T_{Newton}$) that follows Newton's law of cooling, describing its equilibrium tendency with the ambient temperature; and a heat source excitation term ($Q$) expanded by a set of harmonic functions, representing periodic or complex external energy inputs. These two components combine to form a temperature evolution model.

adopted a two-stage strategy: reconstruct 3D structure from RGB images and then map thermal images (Dlesk et al. (2022); Jia et al. (2017); Zhao et al. (2017)). While feasible, these methods underuse thermal information and are restricted to static settings.

Recent works explore neural implicit models. NeRF (Mildenhall et al. (2021)) has been extended to thermal imaging, e.g., ThermoNeRF (Hassan et al. (2024)) and Thermal-NeRF (Ye et al. (2024)), which learn 3D thermal distributions and synthesize novel views. However, implicit NeRFs suffer from unstable geometry and high training/rendering cost, limiting scalability to large or dynamic scenes.

Different from NeRF, 3D Gaussian Splatting (3DGS) (Kerbl et al. (2023)) adopts explicit Gaussians for efficient rendering and editability. Extensions such as Thermal3D-GS (Chen et al. (2024)) and TGA-GS (Zou et al. (2025)) apply 3DGS to thermal modeling, achieving faster rendering but still constrained to static scenes, thus failing to capture temporal temperature evolution.

## 2.2 DYNAMIC RECONSTRUCTION AND THERMODYNAMICS MODELING

To model dynamics, 4DGS (Wu et al. (2024)) extends Gaussians with deformation fields to capture geometry and appearance changes; follow-ups further improve efficiency (Wan et al. (2024); Lee et al. (2024); Zhang et al. (2025); Li et al. (2024c)). Yet, these methods address only appearance, not thermal physical properties, and are unsuitable for thermal dynamic scenes. ThermalGS (Liu et al. (2025)) introduces time embeddings for dynamic thermal modeling, but remains data-driven and prone to physical inconsistency. NTR-Gaussian (Yang et al. (2025)) incorporates thermodynamic parameters via neural prediction and numerical integration, improving physical fidelity but at high computational cost due to its implicit representation.

In contrast, our method directly embeds thermophysical parameters into Gaussian primitives and derives closed-form, differentiable time solutions. This design avoids numerical integration, maintains efficiency comparable to static 3DGS, and yields physically consistent dynamic thermal reconstructions.

## 3 METHOD

We present Explicit Thermal Gaussian Splatting (ETGS), which embeds thermophysical modeling into an explicit Gaussian representation. Each Gaussian carries geometry and a time-evolving temperature state governed by two components (Fig. 2): an exponential term from Newton's cooling toward ambient equilibrium, and a heat source excitation expanded on harmonic bases to capture

periodic/complex inputs. Their combination yields a closed-form, differentiable temperature evolution that can be evaluated at any timestamp—without numerical integration or implicit neural regression—thereby naturally handling unequal and out-of-order sampling. This design attains training/rendering efficiency close to static 3DGS while accurately reconstructing fast thermal dynamics, striking a balance among accuracy, efficiency, and physical plausibility.

## 3.1 THERMAL GAUSSIAN FIELD

In 3DGS, a scene is an explicit Gaussian field of millions of parameterized Gaussians $G_i$. Its basic form is:

$$G_i = \{\mu_i, \Sigma_i, R_i, \alpha_i, f_i\}, \tag{1}$$

where $\mu_i \in \mathbb{R}^3$ represents the center position of Gaussians, $\Sigma_i$ represents the covariance matrix, $R_i$ represents the rotation matrix, $\alpha_i$ represents the opacity, and $f_i$ represents the characteristic coefficient related to color or radiation (represented in the form of spherical harmonic function expansion). This explicit representation makes 3DGS efficient and flexible in geometry and appearance modeling. However, for thermal scenes, the various properties of the Gaussians need to be redefined. As the determining factor of thermal radiation, temperature is necessary to be explicitly represented in modeling. To this end, we expand the Gaussians $G_i$ into thermal Gaussians $\widetilde{G}_i$ and remove the optical properties (spherical harmonic color) to focus on thermal reconstruction. The definition of thermal Gaussians is:

$$\widetilde{G}_i = \{\mu_i, \Sigma_i, R_i, \alpha_i, C_i, h_i, Q_i(t), T_i(t)\} . \tag{2}$$

New thermal properties include: equivalent heat capacity $C_i$, which characterizes the inertia of the Gaussians to temperature changes and determines how quickly the temperature responds to external stimuli; heat transfer coefficient $h_i$, which describes the rate of heat exchange between the Gaussian body and the environment; heat source excitation $Q_i(t)$, which represents the energy input to the thermal Gaussians over time and is expanded using a set of Fourier basis functions to characterize complex or periodic thermal processes; and temperature state $T_i(t)$, which represents the temperature of the thermal Gaussians at time $t$ and is analytically solved by a thermodynamic evolution model (see Section 3.2). This design preserves the explicit controllability of the scene and lays the foundation for physically consistent and efficient rendering of subsequent dynamic thermal processes.

## 3.2 THERMODYNAMIC EVOLUTION

This section details the thermodynamic evolution of thermal Gaussians, including the first-order linear differential equations and their analytical solutions, derived from Newtonian heat exchange and heat source excitation, and the closed-form solution obtained after expanding the heat source $Q_i(t)$ over a global frequency grid using a harmonic basis.

**Continuous ODE model.** For the $i$-th thermal Gaussian, assume the environment temperature is a constant $T_{env}$, the equivalent heat capacity $C_i > 0$, the convection/radiation equivalent heat transfer coefficient $h_i \geq 0$, and the heat source is $Q_i(t)$. Based on the energy conservation principle, the first-order linear ODE is obtained:

$$C_i \frac{dT_i(t)}{dt} = -h_i \left(T_i(t) - T_{env}\right) + Q_i(t). \tag{3}$$

Defining the time constant $\tau_i = \frac{C_i}{h_i}$, Eq. 3 can be rewritten as:

$$\frac{dT_i(t)}{dt} = -\frac{1}{\tau_i} \left(T_i(t) - T_{env}\right) + \frac{1}{C_i} Q_i(t). \tag{4}$$

**Analytical solution of the ODE.** Applying the integrating factor method to Eq. 4 (see Appendix A for the detailed derivation) and denoting the initial condition by $T_i(0) = T_{i,0}$, we obtain:

$$T_i(t) = T_{env} + \left(T_{i,0} - T_{env}\right) e^{-\frac{t}{\tau_i}} + \frac{1}{C_i} \int_0^t e^{-\frac{t-s}{\tau_i}} Q_i(s) ds. \tag{5}$$

**Harmonic expansion of the heat source.** We expand the heat source on a globally shared frequency grid:

$$Q_i(t) = \sum_{k=1}^{K} A_{i,k} \sin\left(\omega_k t\right) + B_{i,k} \cos\left(\omega_k t\right), \tag{6}$$

where $\{\omega_k\}_{k=1}^K$ are sampled from a log-uniform frequency grid:

$$\omega_k = \omega_{min} \left( \frac{\omega_{max}}{\omega_{min}} \right)^{\frac{k-1}{K-1}}, \ k = 1, \dots, K. \tag{7}$$

The bounds $[\omega_{\min}, \omega_{\max}]$ and the number of frequencies $K$ are jointly determined by the sampling geometry (observation duration $T_{\text{span}}$, minimum sampling interval $dt_{\min}$) and the thermodynamic priors ($\tau_{\min}$, steady-state gain threshold $\alpha$) (see Appendix B for construction details).

**Closed-form solution of the ODE.** Substituting Eq. 6 into Eq. 5 and evaluating the kernel integrals term by term (see Appendix A for the detailed derivation), we obtain a closed-form expression for the temperature of each thermal Gaussian at arbitrary time $t$:

$$\begin{aligned}
T_i(t) = {} & T_{\text{env}} + \left( T_{i,0} - T_{\text{env}} \right) e^{-t/\tau_i} \\
& + \sum_{k=1}^K \frac{\tau_i}{C_i \left( 1 + (\omega_k \tau_i)^2 \right)} \Big\{ A_{i,k} \big[ \sin(\omega_k t) - \omega_k \tau_i \cos(\omega_k t) + \omega_k \tau_i \, e^{-t/\tau_i} \big] \\
& \qquad\qquad + B_{i,k} \big[ \cos(\omega_k t) + \omega_k \tau_i \sin(\omega_k t) - e^{-t/\tau_i} \big] \Big\}.
\end{aligned} \tag{8}$$

### 3.3 Dynamic Thermal Rendering

During dynamic thermal rendering, temperatures need be mapped to colors. We adopt a linear model for this mapping. Specifically, we linearly normalize temperatures to grayscale using the temperature bounds measured during acquisition. Let the scene's temperature range be $[T_{\min}, T_{\max}]$ and the temperature of the $i$-th thermal Gaussian at time $t$ be $T_i(t)$. The corresponding grayscale intensity is:

$$I_i(t) = clip \left( \frac{T_i(t) - T_{min}}{T_{max} - T_{min}}, 0, 1 \right), \tag{9}$$

where $\text{clip}(\cdot, 0, 1)$ truncates values to $[0, 1]$. During training, we use continuous intensity ($[0, 1]$ interval) to participate in differentiable loss, and then map it to pseudo color during visualization to enhance the visual effect.

For rendering, we perform standard alpha compositing along each ray while keeping the usual transmittance $Tr_i$ and opacity $\alpha_i$, but use $I_i(t)$ instead of the SH color term of 3DGS. The dynamic thermal rendering follows the equation below:

$$C = \sum_{i=1}^N Tr_i \alpha_i I_i(t). \tag{10}$$

### 3.4 Training and Optimization

Our overall training and optimization framework follows the traditional 3DGS optimization process, using the difference between the rendered image and the ground-truth image to drive gradient back-propagation, thereby continuously updating the geometric and thermal properties of the Gaussians. Based on the original 3DGS loss function, we introduce an additional regularization constraint based on the characteristics of dynamic thermal scenarios to stabilize the learning of the heat source harmonic parameters $A_{i,k}$ and $B_{i,k}$ to prevent overfitting and oscillation. The final total loss function is:

$$\mathcal{L}_{total} = (1 - \lambda)\,\mathcal{L}_1 + \lambda \mathcal{L}_{D-SSIM} + \lambda_{reg} \sum_{i,k} \left( A_{i,k}^2 + B_{i,k}^2 \right), \tag{11}$$

Where $\mathcal{L}_1$ and $\mathcal{L}_{D-SSIM}$ are widely used in Nerf/3DGS-style reconstruction (Kerbl et al. (2023)), their specific definitions can be found in Appendix E. This loss design allows our method to preserve rendering accuracy while mitigating overfitting in thermal dynamics modeling, thereby ensuring robust convergence even in complex dynamic scenes.

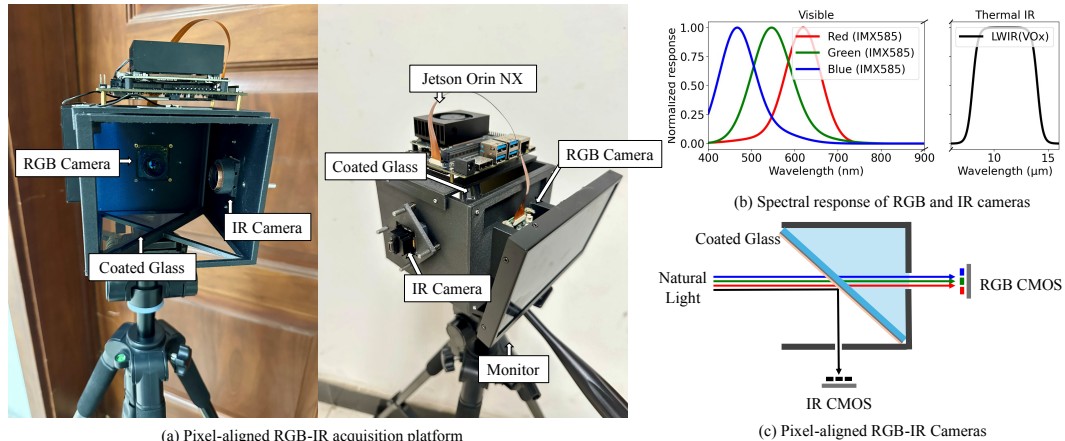

(a) Pixel-aligned RGB-IR acquisition platform

(b) Spectral response of RGB and IR cameras

(c) Pixel-aligned RGB-IR Cameras

Figure 3: **Pixel-aligned RGB-IR acquisition platform.** (a) All devices are mounted on a rigid frame to ensure stability. (b) Response bands of the RGB and IR cameras: The RGB camera uses the IMX585 chip (response band 300-800nm), and the IR camera uses an uncooled vanadium oxide (VOx) microbolometer sensor (response band 8-14μm). (c) Optical principle of pixel-level alignment: A piece of coated glass (coating materials: zinc sulfide, silver) is mounted within the black frame. Visible light passes through the glass and enters the front RGB CMOS, while infrared light is reflected by the glass and reaches the side IR CMOS. The two imaging paths share the same incident light beam, which is split into different wavelengths at the beam splitter.

## 4 RHD: RAPID HEAT DYNAMICS DATASET

### 4.1 PIXEL-ALIGNED RGB-IR ACQUISITION PLATFORM.

To efficiently and economically acquire dynamic thermal scene data, we designed and constructed a pixel-level aligned RGB-IR acquisition platform, as shown in Fig. 3. The core optomechanical structure consists of a piece of selectively coated glass (coating materials: zinc sulfide, silver) positioned at a 45-degree angle. Visible light is transmitted to the front-facing RGB camera, while infrared light is reflected to the side-facing infrared camera, achieving coaxial imaging and zero-baseline beam splitting. The two images are time-stamped and synchronously captured by the Jetson Orin NX. For calibration and alignment, we first perform intrinsic distortion correction on each camera. We then estimate pixel-level alignment error on a common checkboard.

### 4.2 DATASET CONSTRUCTION.

We introduce a new dataset called Rapid Heat Dynamics (RHD), acquired using the pixel-aligned RGB-IR acquisition platform described in section 4.1. The basic specifications of the RHD dataset include: 512×410 resolution, 10 dynamic thermal scenes, and a total of 2363 views. The RHD dataset covers dynamic thermal scenes with rapid temperature changes associated with typical thermodynamic processes, such as cooling, warming, heating, and heat transfer; covers a variety of materials, including metals, fabrics, and organic materials; and covers temperatures ranging from low to high temperatures (-1.0°C to 101.0°C). It also includes millisecond-accurate timestamps. For each scene, we provide pixel-level aligned RGB, Thermal (original), and Thermal (pseudo) images, as well as scene metadata (number of views, temperature range, time range, and ambient temperature), to facilitate both algorithm use and human interpretation. A detailed description of each scene is included in Appendix C. RHD focuses on both cross-modal geometric consistency and thermal dynamic richness, providing a high-quality benchmark for subsequent multimodal research, dynamic thermal scene rendering, and physical prior learning.

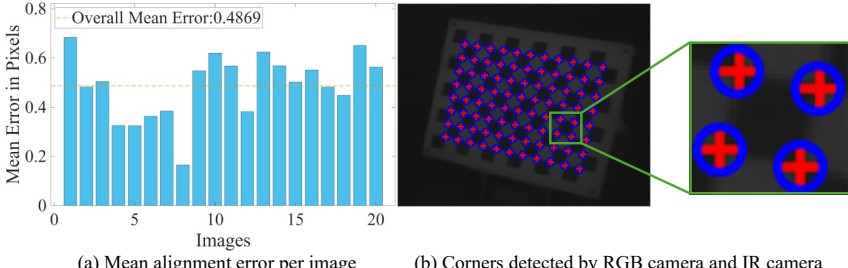

(a) Mean alignment error per image    (b) Corners detected by RGB camera and IR camera

Figure 4: **Alignment Error Verification.** (a) Average alignment error for each of the 20 RGB-IR image pairs. The overall average error is 0.4869 pixels, reaching sub-pixel accuracy. (b) Comparison of checkboard corner detection results. The RGB camera corners (blue circles) and the IR camera corners (red crosses) almost coincide.

## 5 EXPERIMENTS

### 5.1 IMPLEMENTATION DETAILS.

We adopt 3DGS as the backbone, with all experimental settings kept consistent with the original 3DGS framework. Models are trained for 30k iterations under the same hyperparameters, with the specific regularization weight set to $\lambda_{\text{reg}} = 1 \times 10^{-5}$. All experiments share the same train/test split, as well as identical initial point clouds and camera poses. We use RGB images to obtain the initial point clouds and camera poses. During training, we use the raw grayscale thermal images as ground truth for differentiable loss computation; for visualization, the grayscale images are mapped to pseudocolor to enhance perceptual clarity.

### 5.2 ALIGNMENT ERROR VERIFICATION

To verify the alignment error of the constructed pixel-aligned RGB-IR acquisition platform, we first perform intrinsic distortion correction on both the RGB and IR cameras. Then estimate the alignment error on a same checkboard. We compare the corner coordinates $x_{RGB}$ detected by the RGB camera with the corner coordinates $x_{IR}$ detected by the IR camera to accurately calculate the alignment error. The alignment error $\varepsilon_{align}$ is defined as:

$$\varepsilon_{align} = ||x_{IR} - x_{RGB}||_2 . \tag{12}$$

As shown in Fig. 4, we captured 20 pairs of RGB-IR images for evaluation, with an overall alignment error of 0.4869 pixels, reaching sub-pixel accuracy.

### 5.3 RESULTS AND EVALUATION

We compare the proposed method with the state-of-the-art static scene reconstruction methods (3DGS, Mip-Splatting and Thermal3D-GS) and dynamic scene reconstruction methods (4DGS and NTR-Gaussian). We quantitatively evaluate the reconstruction quality based on three metrics (PSNR (Hore & Ziou (2010)), SSIM (Wang et al. (2004)), LPIPS (Zhang et al. (2018)).

Tab. 1 summarizes the comparison of our method with the five baselines. As can be seen, our ETGS achieves the best results in all metrics, significantly outperforming the existing methods. This improvement mainly comes from two aspects: First, compared with the static reconstruction baseline, ETGS explicitly introduces the heat exchange process into the model, which can characterize rapid heating/cooling and subtle heat transfer phenomena between objects, which static models cannot do. Second, our closed-form thermodynamic evolution formula can directly calculate the temperature at any time, avoiding the error accumulation and fuzzification caused by numerical integration, thereby maintaining temporal consistency. In comparison with NTR-Gaussian, we achieve a PSNR improvement of approximately 5 dB, demonstrating the advantage of closed-form thermodynamic modeling in capturing dynamic processes.

Table 1: Quantitative evaluation of our method compared to previous work.

| Metric | Method | Cooling Checkboard | Warming Bottles | Cooling Dumbbells | Cooling Bench | Cooling Ebike | Heat Transfer | Warming Peaches | Heating Workpieces | Warming Cups | Warming Workpieces | Avg. |
|---|---|---|---|---|---|---|---|---|---|---|---|---|
| PSNR↑ | 3DGS | 36.27 | 32.37 | 29.78 | 29.74 | 34.66 | 39.47 | 30.48 | 34.47 | 24.43 | 29.94 | 32.16 |
| | Mip-Splatting | 36.48 | 32.44 | 28.78 | 29.13 | 35.16 | 37.45 | 32.74 | 30.06 | 26.18 | 26.67 | 31.51 |
| | Thermal3D-GS | 39.47 | 33.61 | 32.89 | 32.11 | 38.77 | 39.83 | 34.12 | 34.93 | 30.96 | 30.06 | 34.68 |
| | 4DGS | 33.58 | 30.52 | 34.50 | 35.61 | 33.99 | 39.75 | 34.64 | 30.36 | 34.15 | 32.34 | 33.94 |
| | NTR-Gaussian | 41.82 | 34.63 | 35.18 | 35.56 | 38.09 | 38.99 | 32.44 | 33.65 | 28.18 | 31.01 | 34.96 |
| | Ours | **44.73** | **40.23** | **37.63** | **38.25** | **41.67** | **42.27** | **41.78** | **36.70** | **39.41** | **44.16** | **40.68** |
| SSIM↑ | 3DGS | 0.978 | 0.988 | 0.968 | 0.960 | 0.980 | 0.990 | 0.987 | 0.988 | 0.968 | 0.974 | 0.978 |
| | Mip-Splatting | 0.974 | 0.988 | 0.961 | 0.958 | 0.980 | 0.986 | 0.988 | 0.979 | 0.972 | 0.969 | 0.976 |
| | Thermal3D-GS | 0.983 | 0.989 | 0.981 | 0.970 | 0.983 | 0.990 | 0.990 | 0.989 | 0.978 | 0.976 | 0.983 |
| | 4DGS | 0.963 | 0.979 | 0.976 | 0.967 | 0.962 | 0.986 | 0.984 | 0.960 | 0.975 | 0.970 | 0.972 |
| | NTR-Gaussian | 0.986 | 0.987 | 0.980 | 0.974 | 0.974 | 0.987 | 0.984 | 0.982 | 0.974 | 0.978 | 0.981 |
| | Ours | **0.987** | **0.992** | **0.989** | **0.983** | **0.985** | **0.991** | **0.994** | **0.990** | **0.989** | **0.994** | **0.989** |
| LPIPS↓ | 3DGS | 0.066 | 0.087 | 0.072 | 0.075 | 0.031 | 0.057 | 0.091 | 0.031 | 0.131 | 0.137 | 0.078 |
| | Mip-Splatting | 0.068 | 0.080 | 0.093 | 0.078 | 0.031 | 0.075 | 0.073 | 0.043 | 0.126 | 0.178 | 0.085 |
| | Thermal3D-GS | 0.060 | 0.095 | 0.050 | 0.063 | 0.029 | 0.062 | 0.092 | 0.030 | 0.110 | 0.128 | 0.072 |
| | 4DGS | 0.101 | 0.075 | 0.047 | 0.076 | 0.065 | 0.056 | 0.064 | 0.073 | 0.105 | 0.099 | 0.076 |
| | NTR-Gaussian | 0.072 | 0.092 | 0.052 | 0.092 | 0.098 | 0.092 | 0.104 | 0.055 | 0.133 | 0.102 | 0.089 |
| | Ours | **0.054** | **0.072** | **0.030** | **0.050** | **0.028** | **0.054** | **0.060** | **0.028** | **0.085** | **0.042** | **0.050** |

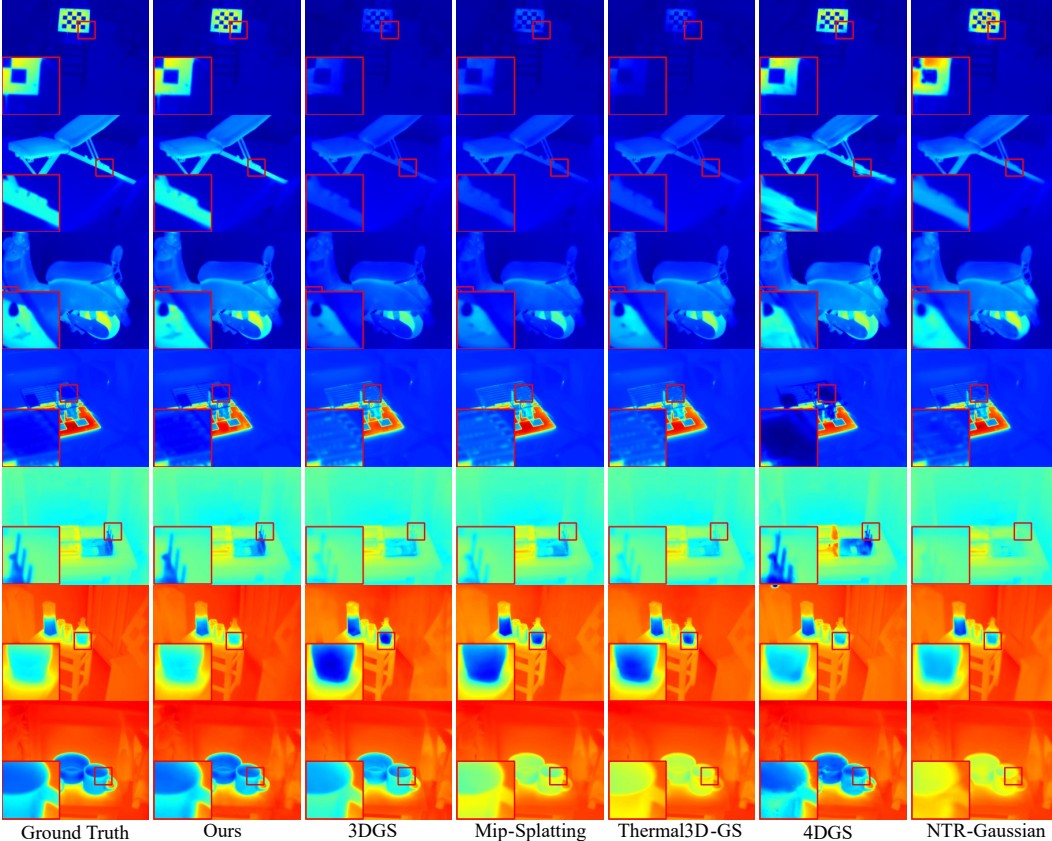

| Ground Truth | Ours | 3DGS | Mip-Splatting | Thermal3D-GS | 4DGS | NTR-Gaussian |

Figure 5: **Comparisons of ours to previous methods.** The scenes are, from the top down:Cooling Checkboard, Cooling Bench, Cooling Ebike, Heating Workpieces, Heat Transfer, Warming Bottles, Warming Cups from the RHD dataset.

Fig. 5 shows a visual comparison of different methods in typical scenes. It can be observed that the static scene reconstruction methods (3DGS, Mip-Splatting and Thermal3D-GS) exhibit significant temperature deviations. This is because these static methods can only learn the average temperature of the scene and cannot obtain accurate temperature values. Dynamic scene reconstruction methods (4DGS and NTR-Gaussian) exhibit varying degrees of artifacts around the edges of some objects.

Table 2: Comparisons of training and rendering efficiency of our method with previous methods.

| Method | Mem (MB)↓ | Time (s)↓ | FPS↑ |
|--------|-----------|-----------|------|
| 3DGS | 2429 | 166 | 557 |
| Mip-Splatting | 2749 | 207 | 589 |
| Thermal3D-GS | 3265 | 470 | 342 |
| 4DGS | 2290 | 1159 | 278 |
| NTR-Gaussian | 4439 | 1469 | 68 |
| Ours | 2391 | 197 | 458 |

Table 3: Ablation Study. We remove the heat source excitation $Q$ and the regularization term separately to evaluate their impact.

| Method | PSNR↑ | SSIM↑ | LPIPS↓ | Mem↓ (MB) | Time↓ (s) | FPS↑ |
|--------|-------|-------|--------|-----------|-----------|------|
| Ours w/o $Q$ | 43.70 | 0.986 | 0.055 | 2385 | 181 | 454 |
| Ours w/o Regular | 42.58 | 0.982 | 0.064 | 2380 | 187 | 463 |
| Ours | **44.73** | **0.987** | **0.054** | 2391 | 197 | 458 |

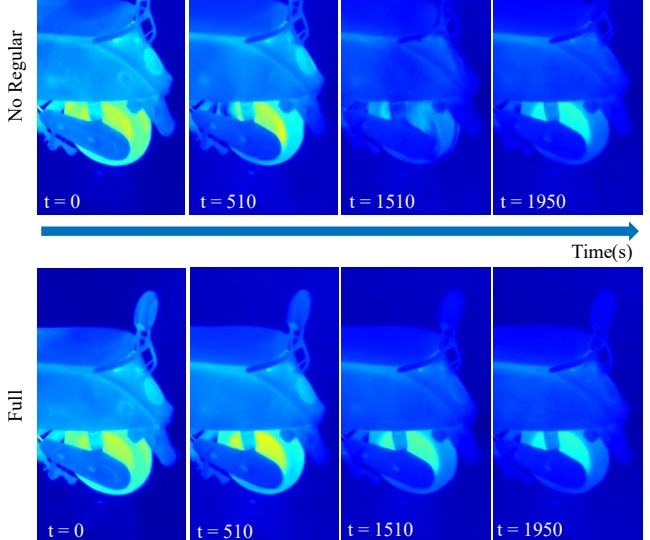

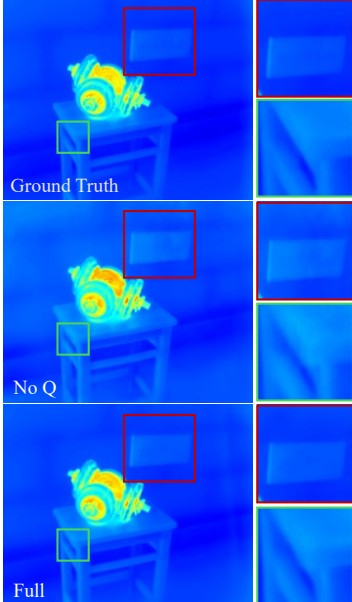

Figure 6: **Visualization of the ablation study.** Left: Ablation of the regularization term. Right: Ablation of the heat source excitation $Q$.

This is because their implicit modeling of time $t$ makes it difficult to ensure temporal consistency. Our method's explicit modeling avoids this drawback, making the model more closely resemble the actual thermal distribution. We provide a detailed visual comparison of all scenes in Appendix F.

Table 4: Ablation Study. Effect of the number of frequencies $K$ on reconstruction quality.

| Metric | K=8 | K=16 | K=24 | K=32 | K=64 |
|--------|-----|------|------|------|------|
| PSNR↑ | 40.57 | 40.59 | 40.68 | 40.77 | 40.95 |
| SSIM↑ | 0.989 | 0.989 | 0.989 | 0.989 | 0.990 |
| LPIPS↓ | 0.050 | 0.051 | 0.050 | 0.051 | 0.051 |

Tab. 2 summarizes average gpu memory, training time, and rendering speed over all scenes. Our method trains in 197 s—close to 3DGS and Mip-Splatting and about an order of magnitude faster than 4DGS and NTR-Gaussian—and renders comparably to static baselines while surpassing implicit methods. This efficiency follows from combining explicit Gaussians with a closed-form temperature solution, avoiding repeated neural-field evaluations. The Appendix D shows the complete calculation cost for ETGS, static baselines and dynamic baselines.

## 5.4 ABLATION STUDY

We removed the heat source excitation $Q$ and the regularization term to analyze their roles in the overall framework. Quantitative results are shown in Tab. 3, and the visualization results are shown in Fig. 6. We evaluated the impact of the number of frequencies $K$ in the heat source excitation $Q$ on the reconstruction results, and the results are shown in Tab. 4.

**Heat source excitation $Q$:** After removing the heat source term, the model can only rely on the exponential decay of Newtonian heat transfer to describe temperature changes, resulting in a lack of external driving force for scene evolution. In this case, the model often fails to capture complex temperature variations or periodic fluctuations, and the learned temperature field exhibits underfitting. As shown on the right side of Fig. 6, after removing $Q$, the model misses fine details and edges in the scene.

**Regularization term:** The regularization term that we introduced constrains the parameters $A_{i,k}$ and $B_{i,k}$ of the frequency basis expansion to prevent them from uncontrolled amplification. After removing this term, although the model still fits the overall trend, unphysical oscillations occur over long time series. This causes the temperature evolution curve to deviate from the smooth physical law, introducing noticeable fluctuation artifacts in the rendering (Fig. 6, left).

**Number of frequencies $K$:** As the number of frequencies $K$ increases, performance improves slightly, but the improvement quickly saturates. Even smaller $K$ (e.g., 8-16) are sufficient to capture the main frequencies, while $K = 24$ strikes a good balance between accuracy and computational cost. Therefore, we used this value in our main experiments. Increasing K above 32 yields very slight improvements (PSNR $< 0.2$ dB, SSIM and LPIPS remain almost unchanged).

## 6 DISCUSSION

**Independent and Coupled Modeling of Gaussians.** Although ETGS models the thermodynamic evolution of each Gaussian as an independent thermodynamic process, the rendered thermal field is not independent between different Gaussians, but rather implicitly coupled. This naturally stems from two mechanisms:(1) overlapping Gaussians. Adjacent Gaussians contribute to the same pixels in the rendered image. Any temperature discontinuity between adjacent Gaussian bodies immediately produces visible artifacts. (2) dense supervision. Infrared images provide hundreds of thousands of constraints for each view. During optimization, the renderer forces Gaussians in local regions to match the same pixel observations, which implicitly achieves smooth temperature changes between adjacent Gaussians. Explicitly modeling heat conduction between Gaussians is an interesting direction for future research. However, it introduces enormous computational and implementation complexity: global coupling between hundreds of thousands of Gaussians, the inability to obtain closed-form solutions, and significantly increased backpropagation costs. ETGS focuses on lightweight, closed-form solutions that are physically accurate and capture the main thermodynamics we observe in controlled dynamic scenes, while also extending explicit inter-Gaussian conduction as an important feature research direction.

**More complex thermodynamic models.** ETGS adopts a first-order linear heat-transfer model consisting of Newtonian cooling and harmonic heat-source excitation. This formulation is intentionally chosen because it admits a closed-form analytical solution, enables efficient optimization comparable to static 3DGS. However, real-world thermodynamic processes may involve nonlinear effects such as temperature-dependent conductivity, radiation coupling, multi-layer material interfaces, or even phase transitions. These phenomena are difficult to incorporate into a closed-form solution, and typically require solving nonlinear PDEs or numerical approximations, which would significantly increase computational cost and compromise the efficiency advantage of explicit Gaussian splatting. Extending ETGS to support nonlinear or higher-order thermal dynamics is an important direction. Developing such models would allow ETGS to generalize to more complex thermal scenarios, including outdoor environments, heterogeneous materials, or strong radiative interactions.

## 7 CONCLUSION

ETGS embeds thermophysics into explicit Gaussians and derives a closed-form temperature solution for efficient, stable dynamic thermal reconstruction. We also release RHD with pixel-aligned RGB-IR and ms-level timing. In future work, we plan to expand RHD to include moving heat sources and more complex environments. Combining RGB with thermal supervision is also a very promising direction to explore.

ACKNOWLEDGMENTS

This work was supported in part by the National Natural Science Foundation of China under Grant 62372235, Grant 62406069, in part by the China Postdoctoral Science Foundation, under Grant 2024M750425.

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

APPENDIX

We organize the appendix as follows:

- **Appendix A:** A complete derivation of the thermodynamic evolution of thermal Gaussians is given (starting from Newtonian heat transfer, obtaining an analytical solution via the integrating factor method, and deriving a closed-form differentiable expression at any time under a harmonic heat source).

- **Appendix B:** Detailed description of the construction principles and implementation of the frequency grid, the determination of upper and lower bounds for sampling and thermal prior constraints, logarithmic sampling and minimum frequency separation, and engineering treatment for robustness.

- **Appendix C:** Report on the scene and time statistics, temperature range, and metadata of the RHD dataset to support reproduction and benchmarking.

- **Appendix D:** The complete calculation cost for ETGS, static baselines and dynamic baselines.

- **Appendix E:** Details about the definitions of common loss functions.

- **Appendix F:** A complete visual comparison of each scene (including local zoom) is provided as an intuitive supplement to the quantitative indicators in the main paper.

- **Appendix G:** Outlook for the potential expansion of the method towards more complex thermal processes and multimodal fusion.

- **Appendix H:** The Use of Large Language Models (LLMs).

## A   COMPLETE DERIVATION OF THERMODYNAMIC EVOLUTION

**Linearization and Integrating Factor:** Rewrite Eq. 3 into standard linear form:

$$\frac{dT_i}{dt} + \frac{1}{\tau_i}T_i = \frac{1}{\tau_i}T_{\text{env}} + \frac{1}{C_i}Q_i(t). \tag{13}$$

Let the integrating factor be $\mu(t) = e^{t/\tau_i}$. Multiplying both sides of the equation by $\mu(t)$ and then differentiating, we obtain:

$$\frac{d}{dt}\left(e^{t/\tau_i}T_i(t)\right) = \frac{1}{\tau_i}T_{\text{env}}e^{t/\tau_i} + \frac{1}{C_i}Q_i(t)\,e^{t/\tau_i}. \tag{14}$$

Integrating over $t \in [0, t]$ and denoting the initial condition $T_i(0) = T_{i,0}$, then obtain:

$$e^{t/\tau_i}T_i(t) - T_{i,0} = \frac{1}{\tau_i}T_{\text{env}}\int_0^t e^{s/\tau_i}\,ds + \frac{1}{C_i}\int_0^t e^{s/\tau_i}Q_i(s)\,ds, \tag{15}$$

where multiplying both sides by $e^{-t/\tau_i}$ gives:

$$T_i(t) = T_{\text{env}} + (T_{i,0} - T_{\text{env}})\,e^{-t/\tau_i} + \frac{1}{C_i}\int_0^t e^{-(t-s)/\tau_i}Q_i(s)\,ds. \tag{16}$$

Eq. 16 is the same as Eq. 5 in the main text.

**Heat source and kernel convolution:** Eq. 6 in the main text is given as:

$$Q_i(t) = \sum_{k=1}^{K}\left(A_{i,k}\sin\left(\omega_k t\right) + B_{i,k}\cos\left(\omega_k t\right)\right). \tag{17}$$

Two types of kernel convolutions need to be evaluated:

$$I_{sin}(\omega, t) = \int_0^t e^{-(t-s)/\tau_i} \sin(\omega s)\, ds, \tag{18}$$

$$I_{cos}(\omega, t) = \int_0^t e^{-(t-s)/\tau_i} \cos(\omega s)\, ds. \tag{19}$$

Let $a = 1/\tau_i$ and $b = \omega$. Using the indefinite integral identities:

$$\int e^{as}\sin(bs)\, ds = \frac{e^{as}}{a^2 + b^2}\left(a\sin bs - b\cos bs\right), \tag{20}$$

$$\int e^{as}\cos(bs)\, ds = \frac{e^{as}}{a^2 + b^2}\left(a\cos bs + b\sin bs\right). \tag{21}$$

Using $e^{-(t-s)/\tau_i} = e^{-t/\tau_i}e^{s/\tau_i}$ to simplify, we obtain the following definite integrals:

$$I_{sin}(\omega, t) = \frac{1}{a^2 + b^2}\left(\frac{1}{\tau_i}\sin\omega t - \omega\cos\omega t + \omega e^{-t/\tau_i}\right), \tag{22}$$

$$I_{cos}(\omega, t) = \frac{1}{a^2 + b^2}\left(\frac{1}{\tau_i}\cos\omega t + \omega\sin\omega t - \frac{1}{\tau_i}e^{-t/\tau_i}\right). \tag{23}$$

Substituting $a = 1/\tau_i$ and using $a^2 + b^2 = \frac{1+\omega^2\tau_i^2}{\tau_i^2}$ yields the compact forms:

$$I_{sin}(\omega, t) = \frac{\tau_i}{1 + \omega^2\tau_i{}^2}\left(\sin\omega t - \omega\tau_i\cos\omega t + \omega\tau_i e^{-t/\tau_i}\right), \tag{24}$$

$$I_{cos}(\omega, t) = \frac{\tau_i}{1 + \omega^2\tau_i{}^2}\left(\cos\omega t + \omega\tau_i\sin\omega t - e^{-t/\tau_i}\right). \tag{25}$$

Substituting Eq. 24 and Eq. 25 into the convolution term of Eq. 16, we obtain the final closed-form solution:

$$\begin{aligned}
T_i(t) = {}& T_{\text{env}} + \left(T_{i,0} - T_{\text{env}}\right)e^{-t/\tau_i} \\
& + \sum_{k=1}^{K}\frac{\tau_i}{C_i\left(1 + (\omega_k\tau_i)^2\right)}\Big\{A_{i,k}\big[\sin(\omega_k t) - \omega_k\tau_i\cos(\omega_k t) + \omega_k\tau_i\, e^{-t/\tau_i}\big] \\
& \qquad\qquad + B_{i,k}\big[\cos(\omega_k t) + \omega_k\tau_i\sin(\omega_k t) - e^{-t/\tau_i}\big]\Big\},
\end{aligned} \tag{26}$$

which corresponds to Eq. 8 in the main text.

## B  FREQUENCY BOUNDS: JOINT CONSTRAINTS FROM SAMPLING AND THERMODYNAMICS

We expand the heat source $Q_i(t)$ on a log-uniform frequency grid, whose passband $[\omega_{\min}, \omega_{\max}]$ is determined jointly by the observation duration and sampling rate (signal-processing perspective) and thermodynamic time-constant priors (physical perspective), balancing expressiveness and robustness. The initial number of frequencies is set to $K = 24$, and is finally adapted to the number of deduplicated active frequencies, i.e., $K = |\{\omega_k\}|$.

**Lowest frequency $f_{min}$: Determined by the observation duration.** A finite observation duration $T_{span}$ sets the minimum resolvable frequency grid $\Delta f \approx 1/T_{span}$. In other words, if a frequency

is lower than $1/T_{span}$, a full cycle cannot fit within the window and is indistinguishable from even lower components. Hence we set:

$$f_{min} = \max\left(\frac{1}{T_{span}}, 10^{-5}\right). \tag{27}$$

**Sampling upper bound $f_{nyq}$: Constrained by the Nyquist frequency (Shannon (2006)).** For nonuniform timestamps with minimum interval $dt_{min}$, we define an effective maximum sampling rate $f_s^{max} \approx 1/dt_{min}$, which yields a strict anti-aliasing bound:

$$f_{nyq} = \frac{f_s^{max}}{2} \approx \frac{1}{2dt_{min}}. \tag{28}$$

**Thermodynamics upper bound $f_{th}$: Given by the first-order thermal inertia low-pass characteristic.** Model each thermal Gaussian as a first-order linear thermal system with time constant $\tau_i$. Its steady-state amplitude-frequency response to the angular frequency $\omega$ is:

$$|G(j\omega)| = \frac{1}{\sqrt{1 + (\omega\tau)^2}}, \ (\tau > 0) \tag{29}$$

When the frequency increases, the system exhibits a low-pass characteristic and decays at $1/\omega$. We require that the system can observe the heat source at the candidate frequency (the steady-state gain is not less than the threshold $\alpha \in (0, 1]$, then

$$|G(j\omega)| \geq \alpha, \tag{30}$$

equals

$$\omega \leq \frac{1}{\tau}\sqrt{\frac{1}{\alpha^2} - 1}. \tag{31}$$

The corresponding upper limit of the frequency is

$$f_{th} = \frac{1}{2\pi\tau_{min}}\sqrt{\max\left(\frac{1}{\alpha^2} - 1, 0\right)}, \tag{32}$$

where we use the fastest thermal time constant prior $\tau_{min}$ (i.e., the "minimum thermal inertia" of the material/structure) to provide the most permissive physical upper bound. In the special case of choosing the half-power point $\alpha = 1/\sqrt{2}$, the well-known cutoff frequency is obtained as $f_c = 1/(2\pi\tau)$. Moreover, the time constant $\tau$ satisfies $\tau = C/h$, where $C$ is the equivalent heat capacity and $h$ is the heat transfer coefficient in Newton's law of cooling. The exponential decay solution in the time domain and the first-order low-pass behavior in the frequency domain are two consistent characterizations of the same thermal process Bergman (2011).

**Joint frequency constraints:** Combining the sampling and physical constraints, the heat source frequency bounds are set to

$$\omega_{min} = 2\pi f_{min}, \tag{33}$$

$$\omega_{\max} = 2\pi \max(\min(f_{\text{nyq}}, f_{\text{th}}), 1.2\, f_{\min}). \tag{34}$$

We first take $\min(f_{\text{nyq}}, f_{\text{th}})$ to ensure both sampling safety and thermodynamic responsiveness. Then compare this value against $1.2\, f_{\min}$ to guarantee a nondegenerate bandwidth-i.e., even under extremely sparse or short observation windows, the usable band remains at least one frequency-bin wide. This prevents the collapse of the upper and lower bounds to the same point, which would invalidate log-spaced sampling of the frequency grid.

# C  RHD: RAPID HEAT DYNAMICS DATASET

As summarized in Tab. 5 and Tab. 6, we present our RHD dataset. Built with the pixel-aligned RGB-IR acquisition platform of Section 4.1, RHD has a resolution of 512×410, comprises 10 dynamic thermal scenes, and totals 2363 viewpoints with millisecond-accurate timestamps. RHD covers canonical thermodynamic processes-cooling, warming, heating, and heat transfer-and spans diverse materials including metals, fabrics, complex devices, and organic objects, with a temperature range of -1.0 °C to 101.0 °C. For each scene, we provide pixel-aligned RGB, Thermal (raw radiometric grayscale), Thermal (pseudocolor), and scene metadata (number of views, temperature range, time span, ambient temperature). With its rich cross-modal and thermal dynamics content, RHD serves as a high-quality benchmark for multimodal 3D reconstruction, dynamic thermal scene rendering, and learning with physical priors.

Table 5: Each scene of the Rapid Heat Dynamics Dataset.

| Scene | RGB | Thermal(original) | Thermal(pseudo) | Views | Temp. Range (°C) | Time Range (s) | Env. Temp (°C) |
|---|---|---|---|---|---|---|---|
| Cooling Checkboard | | | | 218 | 20.0 72.0 | 2145.488 | 26.5 |
| Cooling Dumbbells | | | | 210 | 28.0 46.0 | 2115.472 | 32.3 |
| Cooling Bench | | | | 221 | 31.0 56.0 | 1689.206 | 32.9 |
| Cooling Ebike | | | | 315 | 26.0 60.0 | 1954.054 | 30.2 |
| Heat Transfer | | | | 250 | 21.0 41.0 | 1730.072 | 26.4 |
| Heating Workpieces | | | | 224 | 12.0 101.0 | 1249.084 | 26.9 |
| Warming Bottles | | | | 209 | 5.0 31.0 | 3988.082 | 26.6 |

Table 6: Each scene of the Rapid Heat Dynamics dataset.

| Scene | RGB | Thermal(original) | Thermal(pseudo) | Views | Temp. Range (°C) | Time Range (s) | Env. Temp (°C) |
|---|---|---|---|---|---|---|---|
| Warming Cups | | | | 224 | -1.0 30.0 | 1678.116 | 26.9 |
| Warming Peaches | | | | 254 | 2.0 30.0 | 2284.465 | 26.9 |
| Warming Workpieces | | | | 238 | 3.0-32.0 | 1554.317 | 26.9 |

Table 7: Complete calculation cost.

| Metric | Method | Cooling Checkboard | Warming Bottles | Cooling Dumbbells | Cooling Bench | Cooling Ebike | Heat Transfer | Warming Peaches | Heating Workpieces | Warming Cups | Warming Workpieces | Avg. |
|---|---|---|---|---|---|---|---|---|---|---|---|---|
| Mem↓(MB) | 3DGS | 2395 | 2313 | 2597 | 2324 | 2674 | 2372 | 2350 | 2340 | 2404 | 2522 | 2429.10 |
| | Mip-Splatting | 3389 | 2215 | 3417 | 2384 | 2694 | 2908 | 2154 | 2384 | 2810 | 3132 | 2748.70 |
| | Thermal3D-GS | 3207 | 3146 | 3497 | 3242 | 3544 | 3118 | 2970 | 3522 | 3154 | 3248 | 3264.80 |
| | 4DGS | 2571 | 1953 | 3043 | 1846 | 3621 | 1846 | 1890 | 2444 | 1794 | 1894 | 2290.20 |
| | NTR-Gaussian | 4544 | 4344 | 4144 | 4130 | 4286 | 4598 | 4490 | 4656 | 4454 | 4740 | 4438.60 |
| | **ours** | 2287 | 2311 | 2367 | 2336 | 2700 | 2408 | 2406 | 2348 | 2384 | 2362 | 2390.90 |
| Train↓(s) | 3DGS | 145 | 177 | 195 | 141 | 166 | 173 | 149 | 160 | 175 | 176 | 165.70 |
| | Mip-Splatting | 215 | 205 | 225 | 187 | 207 | 209 | 193 | 215 | 198 | 217 | 207.10 |
| | Thermal3D-GS | 441 | 438 | 527 | 450 | 454 | 431 | 415 | 522 | 433 | 593 | 470.40 |
| | 4DGS | 791 | 847 | 905 | 1603 | 1521 | 1637 | 1621 | 1054 | 798 | 814 | 1159.10 |
| | NTR-Gaussian | 1741 | 1298 | 1068 | 1025 | 1984 | 1486 | 1374 | 1644 | 1358 | 1711 | 1468.90 |
| | **ours** | 129 | 203 | 279 | 185 | 188 | 188 | 205 | 208 | 186 | 199 | 197.00 |
| FPS↑ | 3DGS | 537 | 550 | 422 | 584 | 562 | 602 | 613 | 544 | 581 | 575 | 557.00 |
| | Mip-Splatting | 630 | 648 | 132 | 606 | 658 | 633 | 643 | 613 | 671 | 652 | 588.60 |
| | Thermal3D-GS | 344 | 342 | 305 | 351 | 351 | 356 | 352 | 312 | 349 | 357 | 341.90 |
| | 4DGS | 284 | 274 | 279 | 284 | 279 | 278 | 283 | 251 | 287 | 281 | 278.00 |
| | NTR-Gaussian | 62 | 67 | 77 | 83 | 85 | 60 | 65 | 58 | 65 | 55 | 67.70 |
| | **ours** | 486 | 442 | 396 | 458 | 449 | 487 | 477 | 421 | 478 | 489 | 458.30 |

## D  COMPLETE CALCULATION COST

Tab. 7 provides a complete comparison of training memory, training time, and rendering FPS across all ten scenes in RHD. ETGS achieves a highly competitive computational profile. In terms of training memory, ETGS requires 2390 MB on average, which is significantly lower than Thermal3D-GS (3265 MB), and dramatically lower than NTR-Gaussian (4439 MB). This demonstrates that the explicit thermodynamic modeling introduces minimal memory overhead compared with static 3DGS, and is far more memory-efficient than dynamic baselines. In terms of training time, ETGS trains in 197 seconds on average, which is much faster than all dynamic baselines (e.g., 4DGS: 1159 s, NTR-Gaussian: 1469 s), and slightly slower than static 3DGS (166 s). In terms of rendering speed, ETGS achieves 458 FPS, which is very close to 3DGS (557 FPS), and far above all dynamic NeRF/3DGS methods. This confirms that the closed-form temperature solution introduces no runtime bottleneck, preserving the hallmark real-time rendering performance of 3DGS. Overall, ETGS maintains the computational efficiency of static 3D Gaussian Splatting while providing physically grounded dynamic thermal modeling.

## E    Definitions of Common Loss Functions

Photometric Loss $\mathcal{L}_1$:

$$\mathcal{L}_1 = \|I_{\text{rendered}} - I_{\text{gt}}\|_1 \,, \tag{35}$$

Where $I_{\text{rendered}}$ represents the rendered image and $I_{\text{gt}}$ represents the ground truth image. This pixel-wise $\mathcal{L}_1$ loss is the standard reconstruction objective used in NeRF/3DGS-style optimization.

D-SSIM (Structural Dissimilarity) Loss:

$$\mathcal{L}_{\text{D-SSIM}} = \frac{1 - \text{SSIM}\left(I_{\text{rendered}}, I_{\text{gt}}\right)}{2}, \tag{36}$$

Where $SSIM$ represents Structural Similarity Index, which is used to measure the similarity between two images in terms of structure, brightness, and contrast (Wang et al. (2004)).

## F    Complete Visual Comparison Results

Fig. 7 shows a visual comparison of all scenes from the RHD dataset. This overall comparison demonstrates the advantages of ETGS in detail fidelity and temporal consistency. 3DGS, Mip-Splatting, and Thermal3D-GS are relatively stable in terms of structural preservation, but they lack the detail and dynamic consistency of temperature gradients, making them unable to reproduce time-varying thermal processes. 4DGS and NTR-Gaussian attempt to model the time dimension, but their implicit modeling makes it difficult to ensure temporal consistency. Our method maintains sharper edges, more reasonable temperature gradients, and dynamic evolution consistent with the actual thermal distribution in all scenes.

## G    Future Work

While ETGS has made significant progress in dynamic thermal scene reconstruction, several areas remain worth exploring.

**Extending the RHD dataset with more complex thermal processes.** While RHD provides millisecond-level RGB–IR observations for a wide variety of thermal processes, future versions of the dataset will incorporate moving or time-varying heat sources, stronger environmental disturbances, and more diverse materials. These additions will enable evaluating thermal reconstruction methods under significantly more challenging real-world conditions.

**Coupled RGB–IR modeling.** A natural extension of ETGS is to integrate a physically informed RGB renderer to jointly model appearance and temperature. Such a coupled model could simulate: temperature-dependent optical effects, including glowing surfaces or emissive materials at high temperatures; appearance changes due to radiative transfer, refraction, or thermal reflection, allowing the RGB channel to reflect thermal variations more faithfully. Integrating such cross-modal interactions would enable richer multimodal reconstruction and enhance applications in monitoring, inspection, and simulation.

**Joint modeling of geometry and temperature evolution.** The current ETGS formulation focuses on dynamic thermal processed under a fixed geometry. A challenging but exciting direction is to extend ETGS to simultaneously model geometric deformation and temperature evolution, potentially by combining our thermodynamic formulation with deformation fields, dynamic Gaussians, or canonical-space warping techniques. This would allow ETGS to handle scenes where both structure and thermal state change over time.

Developing a dynamic thermal reconstruction framework that is both interpretable and generalizable will be a key area of future research.

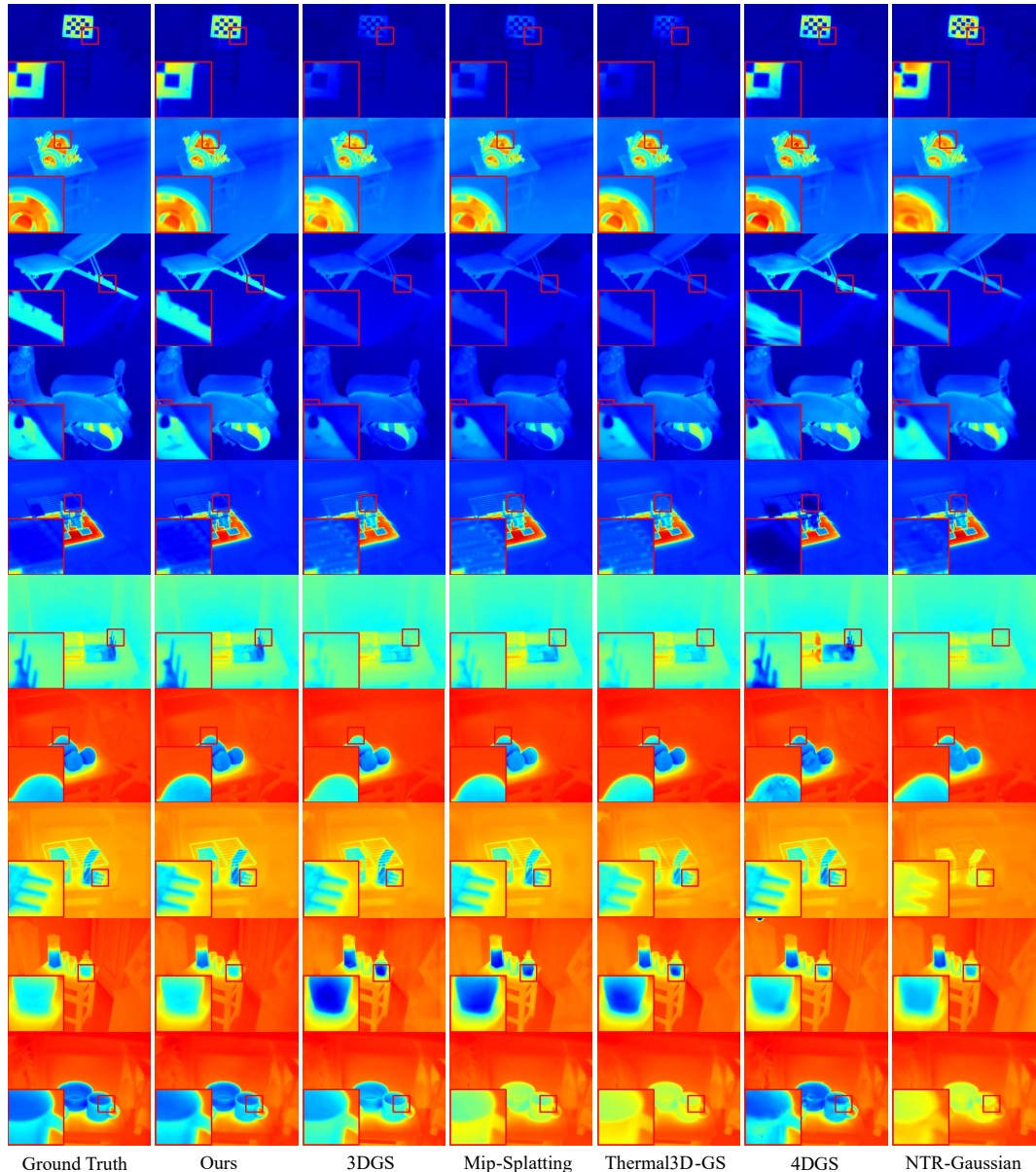

Figure 7: **Comparisons of ours to previous methods on all scenes.** The scenes are, from the top down:Cooling Checkboard, Cooling Dumbbells, Cooling Bench, Cooling Ebike, Heating Workpieces, Heat Transfer, Warming Peaches, Warming Workpieces, Warming Bottles, Warming Cups from the RHD dataset.

## H  THE USE OF LARGE LANGUAGE MODELS (LLMS)

In preparing this manuscript, we employed Large Language Models (LLMs) for language polishing and stylistic refinement, with the goal of improving readability, clarity, and presentation quality.

