# OpenReview forum: "ETGS: Explicit Thermodynamics Gaussian Splatting for Dynamic Thermal Reconstruction"
_ICLR.cc/2026/Conference — ICLR 2026 Poster_

### Official Review · Reviewer_d4dY · 2025-10-24

**Soundness:** 3
**Presentation:** 4
**Contribution:** 3
**Rating:** 8
**Confidence:** 2

**Summary:**

This paper proposes ETGS, a novel extension of 3D Gaussian Splatting (3DGS) that enables the reconstruction of dynamic 3D temperature fields from RGB–IR image pairs at arbitrary viewpoints and time instants.
Unlike conventional 3DGS methods, where each Gaussian stores color information, ETGS instead assigns to each Gaussian a set of parameters that model temporal variations in temperature.
Using these parameters together with a Fourier basis representation, the authors derive an ordinary differential equation (ODE) describing the temporal evolution of temperature and show that its solution can be expressed in a closed form (Eq. (8)). Consequently, ETGS can directly estimate the temperature at any given time using Eq. (8).
The authors also introduce a new dataset, Rapid Heat Dynamics (RHD), which captures thermal phenomena such as warming, cooling, heating, and heat transfer at millisecond temporal resolution using synchronized RGB and IR cameras.
Experimental results on the RHD dataset demonstrate that ETGS outperforms existing 3DGS-based temperature reconstruction methods in terms of accuracy, training efficiency, and rendering speed.

**Strengths:**

* The paper proposes a natural approach to incorporating physically governed variables into the 3DGS framework.

* The experiments demonstrate that the proposed method can effectively model temperature dynamics without compromising the computational efficiency of 3DGS.

**Weaknesses:**

* Although the authors provide an aligned RGB–IR dataset, the proposed model appears to rely solely on IR data and does not utilize RGB information. If RGB input is indeed unnecessary for the model, then constructing an aligned RGB–IR dataset may not be essential. In that case, evaluations such as those shown in Figure 4 could potentially be misleading.

* The only dynamic variable in the current formulation is temperature, while the spatial configuration of objects remains fixed.

**Questions:**

* In Eq. (2), the term $f_i$ from Eq. (1) is omitted. Would it be possible to extend the model to simultaneously render RGB images by retaining $f_i$? It would be sufficient to mention this as a potential direction for future work.

* Regarding Eq. (11), the definitions of $\mathcal{L}\_1$ and $\mathcal{L}_{D-SSIM}$ should be explicitly stated (for example, in the appendix). At the very least, the references from which these loss terms are derived should be clearly cited in the main text.

---

> ### Author Response · Authors · 2025-11-24
> **Response to Reviewer d4dY**
>
> We sincerely appreciate the positive feedback and valuable suggestions provided by the reviewer. Here are our responses to your comments.
>
> **W1: Although the authors provide an aligned RGB–IR dataset, the proposed model appears to rely solely on IR data and does not utilize RGB information. If RGB input is indeed unnecessary for the model, then constructing an aligned RGB–IR dataset may not be essential. In that case, evaluations such as those shown in Figure 4 could potentially be misleading.**
>
> A1: The current ETGS uses IR images as the primary supervisory signal for the thermal field. However, an aligned RGB-IR dataset remains crucial for the following reasons:
>
> - RGB images enable better geometric initialization, especially in thermal regions with less texture, where IR data alone is insufficient and may even fail to initialize.
>
> - This dataset is intended to serve as a benchmark for future methods that may explicitly utilize both RGB and IR data (e.g., joint RGB-IRrendering, cross-modal regularization, or RGB-guided temperature priors).
>
> ETGS currently primarily uses RGB to obtain accurate geometric priors (initial point clouds and camera poses). We elaborate it in section 5.1 "Implementation Details". The RHD dataset is designed to support broader multimodal research and surpassing our current approach.
>
> **W2: The only dynamic variable in the current formulation is temperature, while the spatial configuration of objects remains fixed.**
>
> A2: We thank the reviewer for pointing this out. ETGS was designed to simulate thermodynamics while preserving geometry. Simultaneously recovering the spatiotemporal variations of geometry and temperature is a much more challenging problem: it requires handling non-rigid body motion, topological changes, motion-temperature coupling, and dynamic view-dependent occlusion. ETGS aims to be a first step towards explicit, physically interpretable thermal modeling. We agree that extending ETGS to simultaneously simulate geometric and temperature evolution is an exciting direction for future research, which we plan to explore in future studies. We illustrate this in Appendix G.
>
> **Q1: In Eq. (2), the term $f_i$ from Eq. (1) is omitted. Would it be possible to extend the model to simultaneously render RGB images by retaining $f_i$? It would be sufficient to mention this as a potential direction for future work.**
>
> A1: We thank the reviewer's suggestion. In ETGS, we focused on thermal reconstruction and therefore omitted the characteristic coefficients $f_i$ related to RGB colors. As the reviewer suggested, extending ETGS to simultaneously render RGB and IR images is a promising direction for future work. ETGS provides a physically meaningful thermal field, and incorporating it into RGB rendering can simulate temperature-dependent optical components (e.g., luminescent surfaces at high temperatures) and simulate temperature-affected appearances (e.g., thermal reflection) through radiative transfer or refraction models. Many more derivative works can be developed based on ETGS, which we outline in Appendix G.
>
> **Q2: Regarding Eq. (11), the definitions of $\mathcal{L}_{1}$ and $\mathcal{L}_{D-SSIM}$ should be explicitly stated (for example, in the appendix). At the very least, the references from which these loss terms are derived should be clearly cited in the main text.**
>
> A2: We thank the reviewer for pointing out the missing details. In the revised version, we have clarified the definition of each existing loss term in Appendix E and added the corresponding references in Section "3.4 Training and Optimization".

---

> > ### Comment · Reviewer_d4dY · 2025-11-27
> >
> > Thank you for your response. I now understand that the RGB images are used for the initialization of point clouds and for camera pose estimation. I also agree that having RGB and IR data aligned is important for establishing a useful benchmark dataset in the future. I intend to maintain my current high score for this submission.

---

### Official Review · Reviewer_BcQH · 2025-10-29

**Soundness:** 3
**Presentation:** 3
**Contribution:** 3
**Rating:** 6
**Confidence:** 3

**Summary:**

This paper proposes ETGS (Explicit Thermodynamics Gaussian Splatting), a method for reconstructing dynamic thermal scenes by integrating physics-based thermal modeling into 3D Gaussian Splatting. The approach extends standard 3D Gaussians with thermal properties: equivalent heat capacity, heat transfer coefficient, and time-evolving temperature. Temperature evolution is modeled via a first-order ODE derived from Newton's cooling law plus a heat source term expanded on a harmonic basis using Fourier features. The authors derive a closed-form analytical solution that enables efficient rendering at arbitrary timestamps without numerical integration. The paper also introduces the Rapid Heat Dynamics (RHD) dataset: 10 scenes with 2,410 pixel-aligned RGB-IR views, millisecond-accurate timestamps, and coverage of canonical thermal processes (cooling, warming, heating, heat transfer). Experiments show substantial improvements over baselines while maintaining training efficiency comparable to static 3DGS.

**Strengths:**

- The paper tackles an important and difficult problem at the intersection of computer vision and graphics, with several potential downstream applications.
- The idea of integrating explicit (albeit simplified) thermodynamical modeling into 3DGS is creative, and addresses a real gap in the literature.
- The ODE closed-form solution which avoid numerical integration is elegant and principle
- The model is strongly grounded on physics, enhancing its interpretability
- The dataset introduced in this paper can be of a significant value to the literature. Its temporal resolution, variety of thermodynamic process and adequate hardware design make it a solid dataset with potential reuse in the field and beyond.
- The results are strong, with significant improvements compared to baselines, and with a training cost comparable to static 3DGS, which is much faster than dynamical baselines.
- Evaluation is comprehensive and uses appropriate metrics
- The paper is generally well written and is easy to follow,

**Weaknesses:**

- The thermodynamic model presented in the paper is very simplified, missing or oversimplifying important factors like conduction or radiation, non-linear dynamics, and no phase changes. This undermines the "faithful thermodynamic model" claim. At best, the model is physically plausible for simplified scenarios.
- The usage of Fourier Basis for the harmonic expansion requires further justification. While Fourier approximations make sense, it would be benefitial to understand how different alternatives (eg polynomials, splines, or learned basis) would impact the model. The use of K=24 should be ablated.
- Computational cost should be studied further.
- The paper misses important technical details, including how and why the hyperparameters were chosen.
- There is little validation in term of physics plausibility: For example, the learned h for each material (eg fabric or metal), do they match with known material properties? Is energy conservation preserved? Can it extrapolate beyond training time ranges?
- The dataset has limitations in terms of spatial resolution, and only contains 10 scenes, which limits impact.
- Evaluation is done exclusively on perceptual metrics, missing the temperature accuracy which I argue is as important for this problem.
- The paper is not reproducible in its current state (although code was promised). Many hyperparameter values were not reported.
- Failure cases could be analysed more thoroughly.

**Questions:**

Questions:
- Why is the PNSR improvement across scenes show such a high variance?
- What do the learned A_i,k, B_i,k represent physically?

Other comments:
- Abstract could be written in a less dense way, it lacks conciseness
- Please fix inconsistent notation (eg G_i)

---

> ### Author Response · Authors · 2025-11-24
> **Response to Reviewer BcQH (1/4)**
>
> We sincerely appreciate the positive feedback and valuable suggestions provided by the reviewer. Here are our responses to your comments.
>
> **W1: The thermodynamic model presented in the paper is very simplified, missing or oversimplifying important factors like conduction or radiation, non-linear dynamics, and no phase changes. This undermines the "faithful thermodynamic model" claim. At best, the model is physically plausible for simplified scenarios.**
>
> A1: We appreciate this clarification. We have revised the wording in the "Abstract". Our aim is not to claim a completely general thermodynamic solver, but rather a physically based and interpretable model specifically designed for the dynamic thermal scenes.
>
> **W2: The usage of Fourier Basis for the harmonic expansion requires further justification. While Fourier approximations make sense, it would be benefitial to understand how different alternatives (eg polynomials, splines, or learned basis) would impact the model. The use of K=24 should be ablated.**
>
> A2: We chose the Fourier basis for the following reasons:
> - It provides a general and easily understood basis for representing periodic or smooth time signals, with good approximation properties.
> - The sinusoidal structure fits well with oscillating heating modes (e.g., periodic switching of heat sources, fan-shaped convection).
> - It integrates naturally into the solution of ordinary differential equations, thus obtaining a simple closed form. We acknowledge that other bases (polynomial, spline, or learned basis) are also feasible. However, they either (i) lack the same frequency interpretability or (ii) require additional parameters and regularization to prevent overfitting.
>
> Regarding the choice of $K$, we conducted an ablation experiment with different $K$, demonstrating that the experimental results are not sensitive to the value of $K$. We provide further explanation in Section "5.4 Ablation Study".
>
> |Metric|K=8|K=16|K=24|K=32|K=64|
> |-|-|-|-|-|-|
> |PSNR↑|40.57|40.59|40.68|40.77|40.95
> |SSIM↑|0.989|0.989|0.989|0.989|0.990
> |LPIPS↓|0.050|0.051|0.050|0.051|0.051
>
> **W3: Computational cost should be studied further.**
>
> A3: We appreciate the reviewers' suggestions and have added the single-scene training time, single-frame rendering time, and memory usage of ETGS compared to static and dynamic baselines in Appendix D to further study the computational costs.
>
> **W4: The paper misses important technical details, including how and why the hyperparameters were chosen.**
>
> A4: We appreciate the reviewers' suggestions. For all optimization-related settings (e.g., learning rate, learning rate scheduling, regularization weights, D-SSIM weights, etc.), we followed the same configuration as the baseline to avoid introducing unnecessary degrees of freedom. For the ETGS-specific hyperparameter, the number of frequencies K used to parameterize the heat source term, we conducted ablation experiments, which are reported in Section 5.4 "Ablation Study".

---

> > ### Author Response · Authors · 2025-11-24
> > **Response to Reviewer BcQH (2/4)**
> >
> > **W5: There is little validation in term of physics plausibility: For example, the learned h for each material (eg fabric or metal), do they match with known material properties? Is energy conservation preserved? Can it extrapolate beyond training time ranges?**
> >
> > A5: We appreciate the reviewer's attention to the learned thermophysical parameters. We clarify the following:
> > - **Material properties in the scene are unknown and cannot be measured and verified from images.** Actual heat transfer coefficients and other properties for each material in the scene are unavailable, and the apparent temperature measured by the infrared camera is affected by emissivity, optical properties, and mixed pixel effects. Each Gaussian body typically covers multiple surfaces or composite materials. Therefore, checking for consistency with known material properties is not applicable to image-based reconstruction. Importantly, ETGS itself does not preclude such verification. If future thermal datasets can provide calibrated material properties or controlled settings with known heat transfer coefficients, ETGS can integrate and evaluate this information. We consider this an exciting direction for future research.
> >
> > - **Energy conservation is built into the ordinary differential equation (ODE) formula.** Eq. (3) is directly derived from the law of conservation of energy, and its closed-form solution preserves this condition.
> >
> > - **There are uncertainties in extrapolating outside the training time window.** The thermodynamics in real-world environments may change due to unobserved perturbations, alterations in heating patterns, or variations in boundary conditions. Therefore, long-term extrapolation cannot be reliably verified using image data alone. Similar to existing dynamic scene reconstruction methods such as 4DGS and NTR-Gaussian, our goal is to reconstruct the temperature field over the observation time span, provided that sufficient supervision information is available. Exploring the model's behavior under strong temporal extrapolation is an interesting research direction for the future, but this is beyond the scope of this paper.
> >
> > **W6: The dataset has limitations in terms of spatial resolution, and only contains 10 scenes, which limits impact.**
> >
> > A6: We thank the reviewer's concerns regarding the suitability of the RHD dataset. We would like to clarify:
> >
> > - **The resolution.** The resolution of the RHD dataset depends on the native resolution of research-grade infrared sensors. Most long-wave/mid-wave infrared thermal imagers operate at a resolution of 640×512, which is the standard high-resolution format in infrared imaging. The RHD dataset uses this native resolution with only minor edge cropping to eliminate calibration distortion and improve pixel alignment for RGB-IR correspondence. For ultra-high-resolution infrared sensors (such as the Zihe 1280 thermal imager, with a resolution of 1280x1024 and a price of $4000), its exorbitant price is beyond the reach of our laboratory.
> >
> > - **The number of scenes.** RHD contains 10 fully-aligned, millisecond-level RGB-IR dynamic scenes. This dataset is roughly the same size as the widely used NeRF/3DGS benchmark datasets, many of which contain a comparable number of scenes. For example, Synthetic Blender [1] contains 8 scenes, Mip-NeRF 360 [2] contains 9 scenes, and RGBT-Scenes [3] contains 10 scenes. Compared to these datasets, RHD also provides pixel-aligned multimodal views, rapidly changing dynamic thermal processes, and millisecond-level timestamps. In future work, we plan to expand the RHD dataset to include moving heat sources and more complex environments.
> >
> > [1] Kerbl, Bernhard, et al. "3D Gaussian splatting for real-time radiance field rendering." ACM Trans. Graph. 42.4 (2023): 139-1.
> >
> > [2] Barron, Jonathan T., et al. "Mip-nerf 360: Unbounded anti-aliased neural radiance fields." Proceedings of the IEEE/CVF conference on computer vision and pattern recognition. 2022.
> >
> > [3] Lu, Rongfeng, et al. "Thermalgaussian: Thermal 3d gaussian splatting." arXiv preprint arXiv:2409.07200 (2024).

---

> > > ### Author Response · Authors · 2025-11-24
> > > **Response to Reviewer BcQH (3/4)**
> > >
> > > **W7: Evaluation is done exclusively on perceptual metrics, missing the temperature accuracy which I argue is as important for this problem.**
> > >
> > > A7: We appreciate the reviewer's emphasis on the importance of temperature accuracy. We would like to clarify that evaluating using PSNR/SSIM/LPIPS in the infrared image domain is equivalent to evaluating the accuracy of the reconstructed temperature field, because the infrared image is precisely the measurement that defines the supervisory signal. Any temperature reconstruction error will directly manifest as a difference in the infrared image. Therefore, perceptual and photometric metrics naturally quantify the fidelity of the recovered temperature field relative to the observed data.
> > >
> > > **W8: The paper is not reproducible in its current state (although code was promised). Many hyperparameter values were not reported.**
> > >
> > > A8: The full codes of ETGS will be released after the acceptance, including training codes, preprocessing scripts, configuration files, etc. For all optimization-related settings (e.g., learning rate, learning rate scheduling, regularized weights, D-SSIM weights, etc.), we follow the same configuration as the baseline to avoid introducing unnecessary degrees of freedom. For the ETGS-specific hyperparameter, the number of frequencies K used to parameterize the heat source term, we conducted ablation experiments, which are reported in Section 5.4 "Ablation Study".
> > >
> > > **W9: Failure cases could be analysed more thoroughly.**
> > >
> > > A9: In our experiments with the RHD dataset, ETGS did not exhibit any obvious or systematic failures. Across all ten scenes, the reconstructed thermal fields remained temporally smooth, spatially consistent, and free of noticeable artifacts (Fig. 5). Quantitative results also remained consistently good across all scenes (Tab. 1).

---

> > > > ### Author Response · Authors · 2025-11-24
> > > > **Response to Reviewer BcQH (4/4)**
> > > >
> > > > **Q1: Why is the PNSR improvement across scenes show such a high variance?**
> > > >
> > > > A1: This difference stems from varying scene complexity. For simple environments with small temperature variations, all methods perform well. However, complex environments with significant temperature variations pose a considerable challenge to baseline methods, where ETGS' explicit modeling delivers a greater performance improvement.
> > > >
> > > > **Q2: What do the learned A_i,k, B_i,k represent physically?**
> > > >
> > > > A2: $A_{i,k}$ and $B_{i,k}$ parameterize the effective heat source signal that varies with time for each Gaussian. More precisely, they define the amplitude and phase of each harmonic component in the Fourier expansion of the heat source term. Although they do not correspond to a single physical quantity in a textbook, they collectively represent the intensity and temporal pattern of the Gaussian driven by external/internal heating.
> > > >
> > > > **Q3: Abstract could be written in a less dense way, it lacks conciseness**
> > > >
> > > > A3: We have reorganized and simplified the abstract to more clearly highlight the main contributions.
> > > >
> > > > **Q4: Please fix inconsistent notation (eg G_i)**
> > > >
> > > > A4: We thank the reviewer for pointing out the minor inconsistency in the notation of the Gaussians. In the current draft, $G_i$ represents the fundamental Gaussians, and $\overset{\sim}{G_{i}}$ is used to refer to the thermal Gaussians. We believe this could be clearer, and we have corrected this in Section "3.1 Thermal Gaussian Field".

---

### Official Review · Reviewer_9B6h · 2025-10-31

**Soundness:** 3
**Presentation:** 3
**Contribution:** 2
**Rating:** 4
**Confidence:** 3

**Summary:**

The paper introduces ETGS (Explicit Thermodynamics Gaussian Splatting), a method for dynamic thermal scene reconstruction that embeds explicit thermodynamic equations into Gaussian Splatting. Each Gaussian incorporates physically interpretable parameters—heat capacity, heat-transfer coefficient, heat source, and temperature—with a closed-form solution to the heat ODE that eliminates numerical integration and enables efficient, stable rendering under irregular sampling. The authors also propose the RHD dataset, featuring pixel-aligned RGB–IR pairs with millisecond timestamps. Experimental results demonstrate strong performance and clear advantages over prior methods.

**Strengths:**

* First explicit embedding of thermodynamic laws into Gaussian Splatting.
* Derivation from heat ODE to closed-form analytical solution with physical interpretation.
* State-of-the-art metrics.
* First ms-timestamped RGB-IR benchmark for thermal dynamics.

**Weaknesses:**

* The framework largely relies on established heat-transfer equations and standard ODE solutions, with novelty mainly in integrating these principles into the Gaussian representation rather than introducing new thermodynamic insights.
* The framework is demonstrated on controlled thermal scenes; performance in outdoor or multi-object settings remains unclear.
* Only first-order linear heat transfer is considered; nonlinear effects (e.g., phase change or radiation coupling) are ignored.
* More analysis on frequency-grid resolution ($K$, $\omega_{min}$, $\omega_{max}$) would strengthen robustness claims.

**Questions:**

* Could ETGS extend to coupled RGB + thermal rendering where radiance and temperature interact? This coupling seems crucial for realistic multimodal reconstruction and may reveal new advantages of the proposed explicit thermodynamic formulation.
* How sensitive is the closed-form temperature solution to errors in $\tau_i$ and $h_i$? Would it remain stable for heterogeneous materials?

---

> ### Author Response · Authors · 2025-11-24
> **Response to Reviewer 9B6h (1/2)**
>
> We sincerely appreciate the positive feedback and valuable suggestions provided by the reviewer. Here are our responses to your comments.
>
> **W1: The framework largely relies on established heat-transfer equations and standard ODE solutions, with novelty mainly in integrating these principles into the Gaussian representation rather than introducing new thermodynamic insights.**
>
> A1: We did not introduce any new thermodynamic insights, which we consider the domain of physicists. Our main contribution lies in computer vision/graphics: we are the first to explicitly embed a physically-based thermal model into 3DGS, resulting in a four-dimensional temperature field with a closed-form solution. This significantly improves the quality of reconstruction and rendering while maintaining efficiency comparable to static 3DGS.
>
> **W2: The framework is demonstrated on controlled thermal scenes; performance in outdoor or multi-object settings remains unclear.**
>
> A2: Our current dataset focuses on indoor scenes with static geometry but complex and rapidly changing thermal fields. This type of scene presents significant challenges for dynamic thermal reconstruction and is where we can obtain relatively reliable thermal data. We acknowledge that outdoor and multi-object environments are important application scenarios. As discussed in Appendix G, extending the dataset to more complex environments is an important direction for our future work.
>
> **W3: Only first-order linear heat transfer is considered; nonlinear effects (e.g., phase change or radiation coupling) are ignored.**
>
> A3: The reasons we currently use first-order linear ordinary differential equations are:
> - Our experiments are conducted within a moderate temperature range, where heat convection and heat conduction dominate, and the radiation nonlinearity effect is relatively weak. Furthermore, the IR camera has undergone various processing steps such as radiometric calibration and non-uniformity correction at the factory, further reducing the nonlinear effect.
> - Higher-order or nonlinear partial differential equation models (e.g., radiation coupling, phase transitions) typically cannot obtain simple closed-form solutions, requiring numerical solvers or complex neural network agents, which significantly increases training and rendering costs and weakens one of the core advantages of 3DGS.
>
> Experiments show that even first-order models, when explicitly constructed, significantly outperform previous implicit neural methods while maintaining the efficiency of static 3D Gaussian scattering. We discuss nonlinear effects as a promising but not easy research direction in Section "6 Discussion".
>
> **W4: More analysis on frequency-grid resolution ($K, \omega_{min}, \omega_{max}$) would strengthen robustness claims.**
>
> A4: We thank the reviewers for requesting a more in-depth analysis. We conducted an ablation study on the number of frequencies  $K$, where $K = \{8, 16, 24, 32, 64\}$. The results are shown below:
>
> |Metric|K=8|K=16|K=24|K=32|K=64|
> |-|-|-|-|-|-|
> |PSNR↑|40.57|40.59|40.68|40.77|40.95
> |SSIM↑|0.989|0.989|0.989|0.989|0.990
> |LPIPS↓|0.050|0.051|0.050|0.051|0.051
>
> The results show that performance improves slightly with increasing $K$, but the improvement quickly saturates. Even smaller $K$ ​​(e.g., 8-16) are sufficient to capture the dominant frequencies, while $K=24$ strikes a good balance between accuracy and computational cost. Therefore, we used this value in our main experiments. Increasing $K$ above 32 yields very minor improvements (PSNR < 0.2 dB, SSIM and LPIPS remain almost unchanged).
>
> Regarding the frequency range [$\omega_{min}, \omega_{max}$], this range is automatically calculated based on the sampling conditions of different scenarios; details are provided in Appendix B. The selected range encompasses the dominant frequencies observed in thermodynamics and satisfies a series of sampling laws. As long as this range includes these frequencies, ETGS is not sensitive to small changes in [$\omega_{min}, \omega_{max}$]. We provide further explanation in Section “5.4 Ablation Study.

---

> > ### Author Response · Authors · 2025-11-24
> > **Response to Reviewer 9B6h (2/2)**
> >
> > **Q1: Could ETGS extend to coupled RGB + thermal rendering where radiance and temperature interact? This coupling seems crucial for realistic multimodal reconstruction and may reveal new advantages of the proposed explicit thermodynamic formulation.**
> >
> > A1: Yes, ETGS is the natural basis for this coupling. ETGS has already provided a physically meaningful thermal field. In principle, coupled RGB renderers can simulate:
> >
> > - Temperature-dependent optical components (e.g., luminescent surfaces at high temperatures)
> >
> > - Temperature-affected appearances (e.g., thermal reflections) through radiative transfer or refraction models.
> >
> > We believe this is a promising direction, thanks to our explicit thermodynamic formula. We illustrate this in Appendix G.
> >
> > **Q2: How sensitive is the closed-form temperature solution to errors in $\tau_i$ and $h_i$? Would it remain stable for heterogeneous materials?**
> >
> > A2: (1) The ordinary differential equation used by ETGS is a first-order linear stable system. Its analytical solution consists of a decay exponential term and a bounded harmonic response. As long as $\tau_i, h_i>=0$, the system is essentially stable, and small perturbations of $\tau_i$ or $h_i$ will only result in a proportionally small error in $T_i(t)$. Since the solution is Lipschitz continuous with respect to its parameters, there will be no burst or oscillatory instability. (2) Since $\tau_i$ and $h_i$ are learned directly from dense infrared supervision, ETGS can naturally adapt these parameters to heterogeneous materials, thus maintaining stability.

---

### Official Review · Reviewer_aaeD · 2025-10-31

**Soundness:** 3
**Presentation:** 2
**Contribution:** 2
**Rating:** 4
**Confidence:** 3

**Summary:**

The paper proposes ETGS, a method for **dynamic thermal scene reconstruction** that embeds explicit thermodynamic modeling into the Gaussian Splatting (3DGS) framework. Instead of relying on numerically integrated ODEs, ETGS introduces closed-form temperature evolution equations derived from thermodynamic principles (Newton’s cooling and harmonic heat source excitation).
It also presents a new dataset, RHD (Rapid Heat Dynamics), which includes pixel-aligned RGB–IR image pairs with millisecond timestamps covering heating, cooling, and heat-transfer scenarios. ETGS achieves state-of-the-art reconstruction performance while maintaining efficiency comparable to static 3DGS.

**Strengths:**

1. The explicit thermodynamic modeling is well-motivated and technically sound, enabling elegant combination with 3DGS.
2. The proposed method maintains the merits of 3DGS on training and inference speeds by avoiding costly neural integral inference in prior works.
3. The established RHD dataset is well-designed with pixel-aligned RGB-IR pairs, meaningful for future research.
4. The comparison experiments are comprehensive, including both static and dynamic baselines.

**Weaknesses:**

1. The Gaussians are assumed to be independent of each other in the proposed method, which ignores the real-world heat transfer across nearest Gaussians. Therefore, a video or image demonstration of continuity from Gaussian to Gaussian (or part to part) is helpful.
2. The established dataset (RHD) is relatively small and lacks diversity. Moving heat sources or other factors that influence thermodynamics can be involved to improve the dataset complexity.
3. What is the performance when combining RGB and thermal supervisions?
4. The writing of this paper should be improved before acceptance.

**Questions:**

Please address my concerns in the weakness part.

---

> ### Author Response · Authors · 2025-11-24
> **Response to Reviewer aaeD**
>
> We sincerely appreciate the positive feedback and valuable suggestions provided by the reviewer. Here are our responses to your comments.
>
> **W1: The Gaussians are assumed to be independent of each other in the proposed method, which ignores the real-world heat transfer across nearest Gaussians. Therefore, a video or image demonstration of continuity from Gaussian to Gaussian (or part to part) is helpful.**
>
> A1: This comment is very insightful. While we do treat the temperature evolution of each Gaussian as an independent process, the thermal field is not independent between different Gaussians. In contrast, there is implicit coupling. This naturally stems from two mechanisms:
>
> - Overlapping Gaussians. Adjacent Gaussians contribute to the same pixels in the rendered image. Any temperature discontinuity between adjacent Gaussians immediately produces visible artifacts.
>
> - Dense supervision. The infrared image provides hundreds of thousands of constraints for each view. During optimization, the renderer forces the Gaussiand within local regions to match the same pixel observations, implicitly achieving smooth temperature changes between adjacent Gaussians.
>
> This implicit coupling is directly reflected in our results. The lower left of Fig. 6 already shows the smooth, artifact-free temperature changes over time observed from a fixed viewpoint.
>
> Furthermore, we agree that explicitly modeling heat transfer between Gaussians is an interesting research direction, but it incurs significant computational and implementation overhead (global coupling of hundreds of thousands of Gaussians, inability to obtain closed-form solutions, expensive backpropagation, etc.). We explain this design trade-off in section “6 Discussion”.
>
> **W2: The established dataset (RHD) is relatively small and lacks diversity. Moving heat sources or other factors that influence thermodynamics can be involved to improve the dataset complexity.**
>
> A2: RHD was designed as the first millisecond-level, pixel-aligned benchmark dataset of dynamic thermal scenes with multiple thermal processes. The number of scenes and views is comparable to many NeRF/3DGS class datasets. For example, Synthetic Blender [1] contains 8 scenes, Mip-NeRF 360 [2] contains 9 scenes, and RGBT-Scenes [3] contains 10 scenes. In terms of diversity, the RHD dataset includes:
>
> - a variety of materials (metals, plastics, ceramics, wood, etc.)
>
> - a variety of thermal processes (heating, cooling, external heating, contact heat transfer)
>
> We acknowledge that RHD currently does not cover moving heat sources or more complex environments. This is a limitation of the current hardware setup (the device requires high-precision mounting and calibration, which is difficult to maintain when heat sources move rapidly across the field of view). We plan to expand RHD to include moving heat sources and more complex environments in Appendix G.
>
> [1] Kerbl, Bernhard, et al. "3D Gaussian splatting for real-time radiance field rendering." ACM Trans. Graph. 42.4 (2023): 139-1.
>
> [2] Barron, Jonathan T., et al. "Mip-nerf 360: Unbounded anti-aliased neural radiance fields." Proceedings of the IEEE/CVF conference on computer vision and pattern recognition. 2022.
>
> [3] Lu, Rongfeng, et al. "Thermalgaussian: Thermal 3d gaussian splatting." arXiv preprint arXiv:2409.07200 (2024).
>
> **W3: What is the performance when combining RGB and thermal supervisions?**
>
> A3: Combining RGB with thermal imaging supervisions is a very promising direction, which we have planned in Appendix G. Our current focus is on reconstructing dynamic thermal fields using thermal supervisions , rather than solving the RGB-IR coupling problem. RGB is primarily used to obtain accurate geometric priors (initial point clouds and camera poses), which we further explain in section "5.1 Implementation Details".
>
> **W4: The writing of this paper should be improved before acceptance.**
>
> A4: We appreciate the reviewers' suggestions and have revised the manuscript to make it clearer and easier to understand. We have clarified the definitions of the symbols $G_i$ and $\overset{\sim}{G_{i}}$, and reorganized and simplified the abstract to more clearly highlight the main contributions.

---

### Author Response · Authors · 2025-11-24
**General Comment**

We sincerely thank all the reviewers for their valuable comments and constructive feedback. We are pleased that the reviewers acknowledged the contributions of this paper. We have carefully revised the paper based on the reviewers' suggestions and questions (marked in blue). We have provided detailed responses to each reviewer's questions below.

---

### Meta-Review · Area_Chair_EVbF · 2026-01-02

**Summary:**

This paper proposes ETGS, a novel extension of 3D Gaussian Splatting (3DGS) that incorporates explicit thermodynamic modeling via a first-order ODE with a closed-form solution. The method enables efficient reconstruction and rendering of dynamic temperature fields from infrared (IR) image sequences. The authors also contribute RHD (Rapid Heat Dynamics), a dataset consisting of pixel-aligned RGB–IR views with millisecond-level timestamps, capturing canonical thermal processes.

The reviewers overall recognized the originality of the approach and its potential impact on thermal scene reconstruction. However, several concerns informed the initial borderline assessments:

**Simplified thermodynamic modeling**: Multiple reviewers (9B6h, BcQH) noted that the physics-based model is based on first-order linear ODEs, and does not capture more complex phenomena like nonlinearity, conduction, radiation coupling, or phase change.\
**Dataset limitations**: Reviewers (aaeD, BcQH) found the RHD dataset small and limited in diversity, noting the lack of moving heat sources, outdoor scenes, or multi-object settings.\
**Physics validation**: Questions were raised about the physical meaningfulness of learned parameters (e.g., heat transfer coefficients), whether energy conservation is preserved, and whether the model extrapolates well beyond training time (BcQH).\
**Missing implementation details and reproducibility**: Reviewers (BcQH, 9B6h) requested more details on hyperparameter choices, loss definitions, and computational cost.\
**RGB–IR coupling**: There were questions (aaeD, d4dY) about the role of RGB supervision, and whether the RGB–IR alignment was necessary if ETGS only uses IR during training.\
**Evaluation metrics**: Reviewer BcQH suggested including temperature accuracy or physics-based validation metrics, beyond perceptual metrics like PSNR/SSIM.

**Reviewer Concerns:**

**Addressed Concerns**\
The authors provided a comprehensive and thoughtful rebuttal, and made several clarifications and additions:

***On simplified thermodynamics***: The authors acknowledged the use of linear models and clarified that this is a practical trade-off to enable closed-form solutions and efficient rendering. They also explicitly revised the abstract to avoid overstating the physical fidelity.\
***Dataset limitations***: The authors justified the dataset size by comparing it to standard NeRF datasets and explained the hardware limitations affecting scene diversity. They also committed to expanding RHD in future work.\
***RGB–IR usage***: The authors clarified that RGB images are used for geometric initialization (camera poses, point clouds) and emphasized the value of RGB–IR alignment for future multimodal research.\
***Physics plausibility***: The authors explained that exact material parameters are not accessible from IR images, and that the learned parameters do not directly correspond to textbook values. They emphasized that the ODE model preserves energy conservation, and acknowledged the limitations of extrapolation.\
***Hyperparameters and implementation details***: The authors added ablation studies on frequency basis size (K), clarified that most training settings follow baselines, and promised full code release post-acceptance. They also added loss term definitions and citations.\
***Computational cost***: Appendix D reports training time, rendering time, and memory usage, showing that ETGS maintains efficiency comparable to static 3DGS.\
***Evaluation metrics***: The authors argued that since IR images directly represent temperature, photometric metrics (PSNR, SSIM, LPIPS) are valid proxies for temperature accuracy.

**Remaining or Partially Addressed Concerns**\
***Extrapolation and physical generalization***: While acknowledged as future work, the ability of ETGS to generalize beyond training time ranges or handle nonlinear phenomena remains unexplored.\
***Failure case analysis***: Although the authors report no observable failures, a more in-depth discussion of failure conditions or limitations (e.g., occlusions, noise, calibration errors) would strengthen the presentation.

**Reviewer Scores:**

**Reviewer aaeD (Initial Score: 4)**: Raised concerns about independence of Gaussians, dataset size, RGB usage, and clarity. The authors addressed these with detailed clarifications and added future directions. Likely score: ↑ to 6

**Reviewer 9B6h (Initial Score: 4)**: Questioned novelty (relying on known ODEs), robustness to parameter errors, and potential for RGB-IR coupling. These were convincingly addressed. Likely score: ↑ to 6

**Reviewer BcQH (Initial Score: 6)**: Provided a detailed, balanced review. Raised valid concerns about physics validation, ablations, and reproducibility. The authors addressed most of these, with only extrapolation and advanced physics left for future work. Likely score: ↔ remains 6

**Reviewer d4dY (Initial Score: 8)**: Was enthusiastic and remained supportive after discussion. Likely score: ↔ remains 8

---

### Decision · Program_Chairs · 2026-01-26

Accept (Poster)